# Match Performance of Soccer Teams in the Chinese Super League—Effects of Situational and Environmental Factors

**DOI:** 10.3390/ijerph16214238

**Published:** 2019-11-01

**Authors:** Changjing Zhou, William G. Hopkins, Wanli Mao, Alberto L. Calvo, Hongyou Liu

**Affiliations:** 1Faculty of Physical Activity and Sport Sciences, Polytechnic University of Madrid, Madrid 28040, Spain; zhouchangjing1@gmail.com (C.Z.); alberto.lorenzo@upm.es (A.L.C.); 2College of Sport and Exercise Science, Victoria University, Melbourne VIC 8001, Australia; willthekiwi@gmail.com; 3School of Physical Education & Sports Science, South China Normal University, Guangzhou 510631, China; Mr.mao9393@outlook.com; 4National Demonstration Centre for Experimental Sports Science Education, South China Normal University, Guangzhou 510631, China

**Keywords:** soccer, notational analysis, contextual variable, match performance

## Abstract

To investigate the effects of situational factors (match location, strength of team and opponent) and environmental factors (relative air humidity, temperature and air quality index) on the technical and physical match performance of Chinese Soccer Super League teams (CSL). The generalized mixed modelling was employed to determine the effects by using the data of all 240 matches in the season 2015 collected by Amisco Pro^®^. Increase in the rank difference would increase the number of goal-scoring related, passing and organizing related actions to a small-to-moderate extent (Effect size [ES]: 0.37–0.99). Match location had small positive effects on goal-scoring related, passing and organizing related variables (ES: 0.27–0.51), while a small negative effect on yellow card (ES = −0.35). Increment in relative air humidity and air quality index would only bring trivial or small effects on all the technical performance (ES: −0.06–0.23). Increase in humidity would decrease the physical performance at a small magnitude (ES: −0.55–−0.38). Teams achieved the highest number in the physical performance-related parameters at the temperature between 11.6 and 15.1 °C. In the CSL, situational variables had major effects on the technical performance but trivial effects on the physical performance, on the contrary, environmental factors affected mainly the physical performance but had only trivial or small impact on the technical performance.

## 1. Introduction

Situational factors are the different competitive conditions in which soccer matches are played and may affect the performance of teams and players at a behavioural level [1]. In recent years, intensive research has been conducted to investigate the influence of situational factors, including match location, team and opponent quality, match status/results among others, on soccer match performance [2,3]. Specifically, players and teams playing at home tended to achieve higher numbers in goal scoring, passing and organizing related technical actions, while committing fewer fouls and receiving fewer cards than playing away [2,4]. Players from successful teams generally had more possession-related actions [3,4] and covered more distance including high-speed-running whilst in ball possession [3,5]. Playing against opposition with higher strength demanded a higher level of technical and tactical performance [4,6], as well as higher level of physical performance [7,8]. Winning teams in elite soccer leagues made more shots and shots on goal and performed fewer high-intensity exercises than drawing and losing teams [2,4,9]. 

Environmental factors, such as relative air humidity, temperature and air quality were also identified to affect soccer match performance [3,10,11,12]. The number of sprints performed and the distance covered at high intensity by elite male players in the 2014 FIFA World Cup matches under high environmental stress (at 50% relative humidity, WBGT 28–33 °C or at 75% relative humidity, WBGT 25–29 °C) were significantly lower than under low environmental stress (at 50% relative humidity, WBGT < 24 °C or at 75% relative humidity, WBGT < 20 °C) [11]. A subsequent study analysing the same matches revealed a similar trend and further concluded that the optimum environmental condition for elite male soccer players to perform physical match activity occurred at the temperature of 22 °C and with under 60% relative humidity [10]. Moreover, a one-percent increase in the concentration of particulate matter in ambient air (PM10) would lead to a 0.021% decrease in the number of passes of elite male players in German professional soccer matches (Bundesliga) [12]. The player’s acclimatization and fitness level can also affect the magnitude of environmental influence on the match performance [13,14]. Link and Weber [14] pointed that compare to players from 2. Bundesliga, players from 1. Bundesliga (better players) reduced their total distance to a greater extent when playing in the warm (≥14 °C) than in the neutral (−4 to 13 °C) environments, whilst preserving their ability to undertake the high-intensity activities when required. The teams from Gulf region have more acclimatization to the heat due to their geographical location, the likelihood of favourable outcome (win or draw) for them increased by 3% for every 1-unit increase in temperature difference [13].

Although the above literature provided insights into soccer behaviours under different match circumstances, the effects of situational and environmental factors on soccer match performance were studied separately. Furthermore, most studies have focused on either technical or physical match performance with few combining both categories [15], especially those investigating environmental effects. Hence, it is essential to analyse both the technical and physical match performance taking into account the impact of both situational and environmental factors, so that the pure effect of each element on distinct aspect of performance would be better assessed.

On the other hand, although research in performance analysis of soccer has been developed in depth in the recent years, there is a scarce investigation of Asian soccer, which is comparatively less developed and is in need for more objective feedback [16,17]. Chinese Super League (CSL) would serve as an ideal model as it has attracted huge investment recently and recruited a large number of high-level players and coaches who may have brought in the latest match approaches and tactical concepts [16]. Recent studies on the CSL by Zhou et al. [16] and Mao et al. [17] reported that shots on target, shot accuracy, sprinting distance in ball possession, quality of opposition, number of passes and number of forward passes have an effect on the match outcome. Furthermore, Yang and colleagues [15] found that upper-ranked teams had greater sprinting distance, total distance out of ball possession, possession, possession during the opponent’s half, number of entry passes in the final third of the field and penalty area and 50-50 challenges than did lower-ranked teams. Similarly, Gai et al. [18] examined the performance characteristics of domestic and foreign players based upon playing position in the CSL. Lago-Peñas et al. [19] identified four playing styles in CSL teams (“possession” play, set pieces attack, counterattacking play and transitional play) by examining 20 match-performance indicators. Along with that, due to the extensive and complex topography and geography of China, CSL players have to play in environmental conditions varying from bitter coldness in winter to unbearable heat in summer, from dry season to wet monsoons and from region to region with drastic temperature and humidity differences throughout the year. Moreover, in some cities hosting CSL teams, atmospheric pollution has been very serious with high levels of air quality index. How those could influence match performance of CSL teams and players is to be further investigated and holds a critical applied significance. The aim of the present study was to identify the influence of situational and environmental factors on the technical and physical performance of the CSL soccer teams.

## 2. Materials and Methods 

### 2.1. Sample

Match performance statistics of all 240 matches in the 2015 season of the CSL were analysed. Original data were collected by a semi-automatic computerized video tracking system, Amisco Pro^®^, whose working process, accuracy, validity and reliability have been discussed in detail in prior studies [20,21].

### 2.2. Experimental Approach

In line with the previous literature [17,22], 17 technical performance-related parameters and seven physical performance-related parameters were chosen as dependent variables in the analysis. The grouping and definition of these variables are listed in Table 1. Three situational variables (match location, team strength and opponent strength) and three environmental factors (temperature, humidity and air quality index) were chosen as predictor variables. Environmental data were derived from the China National Environmental Monitoring Centre, which publishes publicly real-time data of weather and air condition. The temperature and humidity were calculated as the average real-time value between kicking-off and ending time of each match inside the stadium, while the air quality index (AQI) was collected from the real-time data closest to the kick-off time of each match from the monitoring station nearest to the match stadium (range of distance: 0.4–6.9 km). AQI is used by government agencies to communicate to the public how polluted the air currently is or how polluted it is forecast to become. The AQI level is based on the level of six atmospheric pollutants, namely sulphur dioxide (SO_2_), nitrogen dioxide (NO_2_), suspended particulates smaller than 10 μm in aerodynamic diameter (PM10), suspended particulates smaller than 2.5 μm in aerodynamic diameter (PM2.5), carbon monoxide (CO) and ozone (O_3_) measured at the monitoring stations throughout each city. Ethics committee approval of this study was gained from the School of Physical Education & Sports Science at South China Normal University [19CTY014].

A generalized mixed linear model was realized with Proc Glimmix in the University Edition of Statistical Analysis System (version SAS Studio 3.6). A random effect for team identity was used to account for repeated measurement on the teams. The fixed effects estimated the effect of situational and environmental factors. Separate Poisson regressions were run in the model taking the value of each of the 17 technical and seven physical performance-related parameters as the dependent variable.

The effect of team strength and opponent strength was estimated by including the difference in the log of the end-of-season ranks as a predictor [23]. Game location was included as a nominal variable with two levels (home and away). Humidity and AQI were included as numeric linear effects using their raw values and their magnitudes were quantified as the effect of two of their standard deviations [24]: the predicted value for a typically high value of the predictor (1SD above the mean) minus that for a typically low value (1SD below the mean). The raw value of temperature was included as a quadratic effect to allow for the possibility and estimation of an optimum temperature defined by the maximum value of the quadratic. If the maximum occurred within the range of environmental temperatures, its confidence limits were derived by parametric bootstrapping [25]. The optimum temperature and confidence limits were reported if at least 90% of the 10,000 bootstrapped samples produced a maximum; otherwise, it was evident that the effect of temperature was approximately linear and was therefore estimated and reported as the effect of two standard deviations.

### 2.3. Statistical Analysis

Uncertainty in the true effects of the predictors was evaluated using non-clinical magnitude-based inference [24] as implemented in the spreadsheet accompanying the package of materials for generalized mixed modelling with SAS Studio [26]. Observed magnitudes and their confidence limits were expressed in standardized units, whereby the difference in means was divided by the observed between-match standard deviation (SD) derived from the mixed model and then evaluated qualitatively with the following scale: <0.2 trivial, 0.2–0.6 small, 0.6–1.2 moderate, 1.2–2.0 large, >2.0 very large. Effects were deemed clear if the 90% confidence interval did not include positive and negative substantial values. Clear effects were reported with a qualitative likelihood that the true effect was either substantial or trivial (whichever probability was greater) using the following scale: <0.5% most unlikely, 0.5–5% very unlikely, 5–25% unlikely, 25–75% possibly, 75–95% likely, 95–99.5% very likely, >99.5% most likely. 

## 3. Results

### 3.1. Descriptive Statistics

Descriptive statistics of all the dependent and independent variables are presented in Table 2.

### 3.2. Effects of Situational Factors

Figure 1 presents the effects of situational variables on the match performance-related variables. As can be seen from the figure, increase in the rank difference (a better team vs. a worse opponent) would substantially increases shot, shot on target, possession, possession in opponent half, pass, pass accuracy, forward pass, forward pass accuracy, opponent 35 m entry, opponent penalty area entry, cross, corner, offside and 50-50 challenge won to a small-to-moderate extent. Meanwhile, it would decrease foul committed, yellow card and red card at a small magnitude. In contrast, change in the rank difference only showed trivial effect on all the seven physical performance-related parameters. Match location (playing at home compared to playing away) had positive small effects on shot, shot on target, possession, possession in opponent half, pass, forward pass, opponent 35 m entry, opponent penalty area entry, cross and corner and a negative small effect on yellow card and trivial effects on the rest of the variables.

### 3.3. Effects of Environmental Factors

Linear effects of humidity and AQI on the technical and physical performance of CSL teams can be found in Figure 2. A two-standard-deviation increment in humidity and AQI would only bring trivial or small effects on all the 17 technical performance-related parameters. The increase in humidity would decrease the seven physical performance-related parameters at a small magnitude. However, a two-standard-deviation increase in AQI would likely bring a small increment in the abovementioned seven physical performance-related parameters. 

Effects of temperature on the match performance are presented in Table 3. The CSL teams had the highest number of shots, forward passes, offsides and fouls committed whilst playing at the temperature of 18 °C, 17 °C, 22 °C and 13 °C, respectively. Teams achieved the most total distance, sprinting distance, sprinting effort, high-speed-running distance, high-speed-running effort, high-intensity-running distance and high-intensity-running effort at the temperature of 11.6 °C, 15.1 °C, 13.2 °C, 12 °C, 10.6 °C, 13.6 °C and 11.6 °C, respectively. While temperature showed trivial or small linear effects on technical performance-related parameters shot on target, possession, possession in opponent half, pass, pass accuracy, forward pass accuracy, opponent 35 m entry, opponent penalty area entry, cross, corner, 50-50 challenge won, yellow card and red card.

## 4. Discussion

This study aimed at identifying the influence of situational and environmental factors on the technical and physical performance of the CSL soccer teams. Our main results include: (i) situational variables (team and opponent’s relative strength, playing at home/away) had major effects on the technical performance but trivial effects on the physical performance; (ii) on the contrary, environmental factors affected mainly the physical performance but had only trivial or small effects on the technical performance. Specifically, we found that an increase in humidity would decrease the physical performance-related parameters with a small magnitude. Nevertheless, an increase in AQI would likely bring a small increment in the physical performance-related parameters and would most unlikely bring a decrease of physical performance. From the quadratic effects of temperature, we could conclude that CSL teams achieved the most shots, forward pass, offside, foul committed, total distance, sprinting distance, sprinting effort, high-speed-running distance, high-speed-running effort, high-intensity-running distance and high-intensity-running effort at temperatures varying from 10.6–22 °C.

### 4.1. Home Advantage

Home advantage in soccer has been discussed in depth and it is believed to affect choice of tactic and strategy in competition [4,8]. Several authors have shown that, in soccer, home teams generally play better than the away side, making more shots, shots on target, performing better in shot accuracy and other offensive performance measures that are closely related to match success, meanwhile achieving fewer match actions and events related to defending [4,27]. A similar advantage in the technical performance for home teams in the CSL was detected in our research as well. Besides, home advantage was also argued as a factor affecting physical performance (high-intensity actions, low-intensity distance covered, total distance covered) in different soccer leagues [7,28,29]. Lago, Casais, Dominguez and Sampaio [29] pointed out that the home teams covered a greater distance than away teams only during low-intensity activity (<14.1 km/h) in the first division of Spanish soccer league. However, Castellano, Blanco-Villasenor and Alvarez [7] demonstrated there is no significant differences for distances covered at different intensities in the same league. Aquino, Munhoz Martins, Palucci Vieira and Menezes [28] found that players perform significantly higher values in max speed, average speed and high-intensity actions in home matches when compared with away matches in the fourth division of Brazilian Championship of soccer. Given that the current results are in accordance with Castellano, Blanco-Villasenor and Alvarez [7]: No meaningful differences were observed in high intensity distance and total distance covered by teams in matches playing at home and playing away. We could hence conclude that home advantage in the CSL only existed in the technical performance but not physical aspects. However, it is worth noting that while the overall physical demands did not vary significantly within the match location, the influence of home advantage in technical and tactical aspects might modify the distribution of physical fitness in teams’ offense and defence in a match, which may warrant further research.

### 4.2. Strength of Team and Opponent

It has been found that, stronger teams were generally more involved in possession-related actions [3,4] and covered more distance and high-speed-running distance whilst in ball possession than weaker teams [3]. Regarding the quality of opponents, previous studies [3,8,30,31] showed that when playing against weaker teams, the stronger team made more attacking related actions (possession, shots, shots on target, crosses, passes, passing accuracy) and less defensive actions (tackles, yellow cards). Our results showed that the technical match performance of CSL teams presented similar trend regarding the quality difference. However, we identified that changes in the rank difference only brought trivial effects on all the seven physical performance-related parameters, which is different from previous results that showed teams covered greater total distance and performed more high-intensity activities when playing against strong opponents than playing against weak opponents [28,32]. This would suggest that physical demands in the matches of CSL were not affected by the difference in the quality of the team and opponent.

### 4.3. Temperature Comfort Zone

Findings from laboratory research showed that high ambient temperature increased the rate at which fatigue in the cardiovascular system and the central nervous system set in References [33,34]. At the other end of the temperature spectrum, low temperatures negatively affected fat and glycogen metabolism [35,36]. These physiological factors mentioned above, which are affected by high and low temperature, impair physical performance to some extent. In other words, there may be a temperature comfort zone to promote athletic performance. Different best temperature comfort zones have been reported in the previous studies. Grantham et al. [37] claimed that the ambient temperature below 22 °C did not pose a heat stress hazard, while temperatures above 22 °C increased the risk of hyperthermia. Chmura et al. [10] showed that the best comfort zone for players to attain high levels of physical activity entailed an air temperature range below 22 °C and a relative humidity range below 60% in 2014 FIFA World Cup Brazil. Link and Weber [14] found a significant decrease in total distance covered by players in soccer matches from neutral (−4 to 13 °C) to warm (≥14 °C) environments in the top German soccer leagues. Therefore, it is logical that soccer players from different countries and competitions may have different best comfort zones of air temperature and humidity as they are living and training in different geographical and climatic conditions. In other words, the size of environmental influence is related with acclimatization and fitness status of players [3,13,14]. In China, the climate differs from region to region because of the country’s extensive and complex topography. Our results tend to reflect that, in general, the CSL teams obtained most shot, forward pass, offside and foul committed at the temperature of 18, 17, 22 and 13 °C, while achieving best physical performance at the temperature between 11.6 and 15.1 °C.

### 4.4. Humidity

We found that an increase in humidity would decrease the physical performance-related parameters with a small magnitude. This is in accordance with the results of Chmura and colleagues [10] which pointed out that high humidity would negatively affect the physical performance by decreasing both total distance covered and distance covered in different intensity zones. Our results already showed that all CSL teams had to play within a very large span of humidity: 12–100%, with an average of over 66%. Past research [38] showed that in high humidity conditions, due to the high moisture content, it was much harder for body heat to be lost by sweat evaporation. It has to divert blood to the skin to increase heat loss from convection and radiation. This would place an additional demand on cardiac output for blood due to the fact that is still required to transport oxygen to the working muscles, so that it will undermine the physical performance. In this research, an increase in humidity would decrease the physical performance-related parameters with a small magnitude but not impair technical performance. But this decrease in physical performance, especially high-intensity actions, was deemed to allow players to maintain a high-level technical performance (pass accuracy, forward pass accuracy) [3,11]. All above-mentioned factors are particular challenges for the soccer coaching staff in their efforts to ensure optimal physical preparation of the players. Coaches need to be fully aware of the important impact of temperature and humidity on physical performance, prescribing appropriate training in advance.

### 4.5. Air Pollution

Last but not least, the unique of this article is considering the impact of air quality on the performance of soccer teams. It is shown that an increase in AQI exhibited only trivial effects on the technical performance, which is different from the findings of Lichter, Pestel and Sommer [12] that showed air pollution had a negative effect on the passes and pass accuracy. Meanwhile, an increase in AQI would likely bring a small increment in the physical performance-related parameters and would most unlikely bring a decrease. At the same time, this result is also contrary to the previous findings [39], which demonstrated that air pollution would decrease the physical performance of soccer players. These divergences can be explained by the difference in the amount of pollutant administered or the nature of exercise protocols. The current finding that the increment of physical performance in high AQI can be interpreted by the fact that the protection against the vascular dysfunction associated with particulate matter (PM) inhalation could potentially improve exercise performance in high-PM conditions [40]. The present finding shows that Chinese soccer players experienced different air quality environments and an increase in AQI did not decrease the soccer players’ acute physical performance. However, Keramidas et al. [41] indicated that although air pollution does not affect the exercise performance but will cause some changes in physiological indicators in their experimental subjects. Therefore, the long-term impact of environmental pollution on the athlete’s well-being is still under caution.

## 5. Conclusions

Our study demonstrated that situational variables had major effects on the technical performance but trivial effects on the physical performance in the CSL. On the contrary, environmental factors affected mainly the physical performance but had only trivial or small effects on the technical performance in the CSL. There may be an ambient temperature comfort zone (10.6–22 °C) to promote soccer performance and higher or lower temperature may impair soccer match performance. An increase in AQI would likely bring a small increment in the physical performance-related parameters and would most unlikely cause a decrease.

Coaching staff could be aware that temperature and humidity would affect the physical performance, while the game location and strength difference would influence the technical performance of football teams. Hence, physical match preparation could consider environmental factors at first, while technical-tactical match preparation could mainly base on situational factors. Given the fact that there may be a temperature comfort zone to achieve the maximum football performance, players could adopt their pacing strategies according to the ambient temperature in match. It can also help the referee set a reasonable water break based on air humidity and temperature.

In the present study, we have not considered the influence of different acclimatization (physiological adaptations) and fitness status of individual players, neither have we investigated the interactive effects of contextual and environmental factors, which could be directions of future research. The AQI values of our study were collected from the air quality monitoring stations nearest to the stadium, which might be not the most accurate real AQI of the playing condition inside the stadium. What is more, our study only focused on the temporary impact of environmental factors on the match performance. Hence, further study could employ more accurate and direct measures to investigate the effects of long-term environmental conditions on the soccer match performance.

## Figures and Tables

**Figure 1 ijerph-16-04238-f001:**
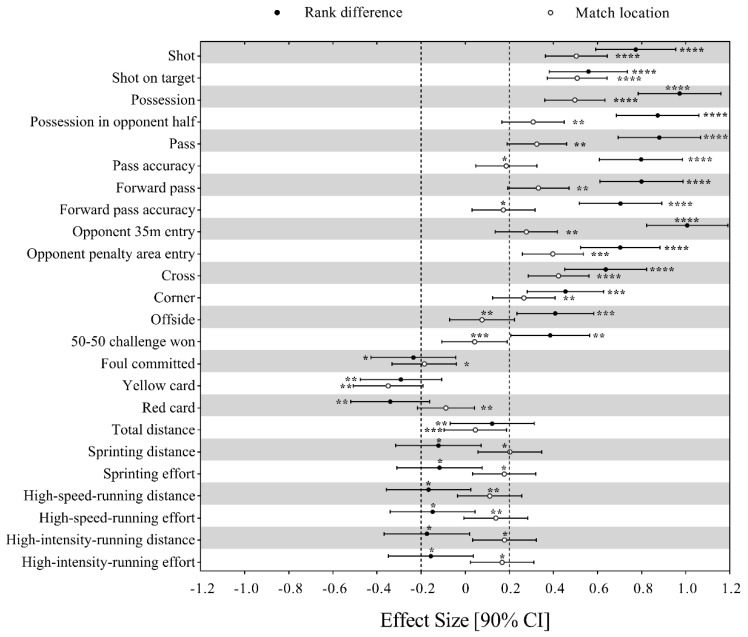
Effects of situational variables on the technical and physical performance of Chinese Soccer Super League (CSL) teams. Effects of team and opponent strength are shown as the effect of an increase of two standard deviations in the value of difference in the log of ranks on the difference of each performance-related parameter. Effects of match location are shown as the effect of playing at home vs. playing away on the difference of each performance-related parameter. Bars are 90% confidence intervals. Dotted lines represent the smallest worthwhile difference. Asterisks indicate the likelihood for the magnitude of the true effect as follows: * possible; ** likely; *** very likely; **** most likely. Asterisks located in the trivial region denote likelihood of trivial effects.

**Figure 2 ijerph-16-04238-f002:**
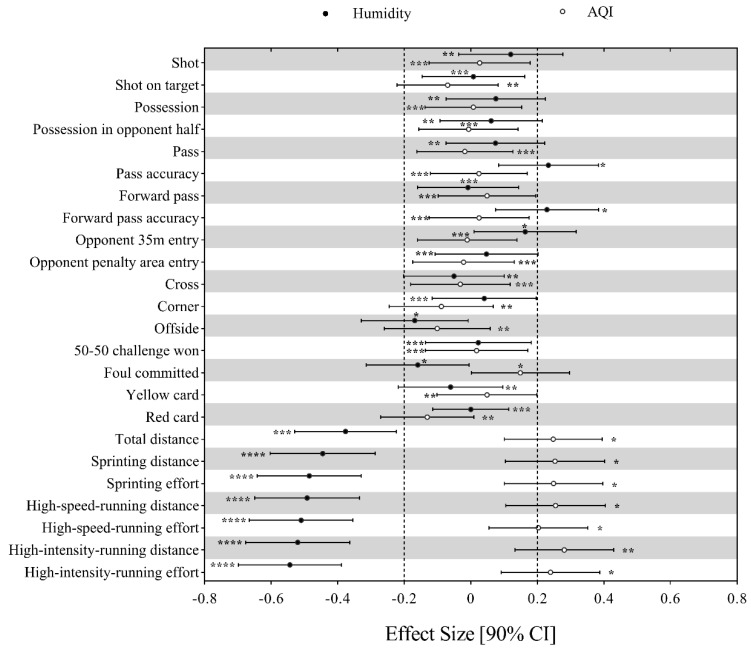
Effects of humidity and AQI (Air quality index) on the technical and physical performance of CSL teams. Effects are shown as the effect of an increase of two standard deviations in the value of humidity and AQI on the difference of each performance-related parameter. Dotted lines represent the smallest worthwhile difference. Asterisks indicate the likelihood for the magnitude of the true effect as follows: * possible; ** likely; *** very likely; **** most likely. Asterisks located in the trivial region denote likelihood of trivial effects.

**Table 1 ijerph-16-04238-t001:** Selected technical and physical performance-related parameters (dependent variables).

Technical Performance-Related Parameters: Operational Definition
**Shot:** an attempt to score a goal, made with any (legal) part of the body, either on or off target **Shot on target:** an attempt to goal which required intervention to stop it going in or resulted in a goal/shot which would go in without being diverted **Possession (%):** the duration when a team takes over the ball from the opposing team without any clear interruption as a proportion of total duration when the ball was in play **Possession in opponent half (%):** possession of a team in opponent’s half of pitch **Pass:** an intentional played ball from one player to another **Pass accuracy (%):** successful passes as a proportion of total passes **Forward pass:** an intentional played ball from one player to another who is located closer to opponent’s goal **Forward pass accuracy (%):** successful forward passes as a proportion of total forward passes **Opponent 35 m entry:** number of times when the ball (possessed by the attacking team) enters the 35 m area (final third of the field) of the opponent’s half of pitch. Each time a player has made an individual possession in the final third of the field, the AMISCO system qualifies it as an opponent 35 m entry of the player who did the individual possession. **Opponent penalty area entry:** number of times when the ball (possessed by the attacking team) enters the penalty area of the opponent’s half of pitch **Cross:** any ball sent into the opposition team’s area from a wide position **Corner:** ball goes out of play for a corner kick **Offside:** being caught in an offside position resulting in a free kick to the opposing team **50-50 challenge won (%):** 50-50% challenge duels won by a team as a proportion of total duels of the match. It is a match action when two players are competing for a ball. A 50-50 challenge must have the following characteristics: Ball is not in control by any player. The two players have roughly a 50-50% chance of gaining control of the ball. Starts when the two players make an attempt to get the ball. Ends when one of the two players touches the ball & the other competing player stops making an attempt to get the ball. **Foul committed:** any infringement that is penalized as foul play by a referee **Yellow card:** where a player was shown a yellow card by the referee for reasons of foul, persistent infringement, hand ball, dangerous play, time wasting and so forth. **Red card:** where a player was sanctioned a red card by the referee, including straight red card and a red card from the second yellow card.
**Physical; Performance-Related Parameters: Operational Definition**
**Total distance (km):** distance covered in a match by all the players of a team **Sprinting distance (km):** distance covered at the speed over 23 km/h in a match by all the players of a team **Sprinting effort:** number of sprinting in a match by all the players of a team **High-speed-running distance (km):** distance covered at the speed of 19.1–23 km/h in a match by all the players of a team **High-speed-running effort:** number of high-speed-running in a match by all the players of a team **High-intensity-running distance (km):** distance covered at the speed over 19 km/h in a match by all the players of a team **High-intensity-running effort:** number of high-intensity-running in a match by all the players of a team

**Table 2 ijerph-16-04238-t002:** Descriptive statistics of all the analysed variables.

Dependent Variables	*n*	Mean	SD	Min.	Max.
Shot	478	12.3	4.9	1	33
Shot on target	478	4.6	2.7	0	16
Possession (%)	478	50.0	7.4	31.0	69.0
Possession in opponent half (%)	478	44.3	7.5	21.0	64.0
Pass	478	363	95	143	687
Pass accuracy (%)	478	79.6	5.7	52.0	92.0
Forward pass	478	123	25	49	202
Forward pass accuracy (%)	478	63.8	8.2	34.0	94.0
Opponent 35 m entry	478	44	14	14	94
Opponent penalty area entry	478	6.9	3.8	0	24
Cross	478	14.5	6.6	2	40
Corner	478	4.6	2.8	0	16
Offside	478	2.3	1.8	0	8
50-50 challenge won (%)	478	50.0	6.5	29.0	71.0
Foul committed	478	17.1	5.1	4	33
Yellow card	478	1.9	1.4	0	6
Red card	478	0.07	0.28	0	3
Total distance (km)	478	109.5	4.9	91.1	122.3
Sprinting distance (km)	478	2.11	0.46	1.1	3.7
Sprinting effort	478	100	20	54	171
High-speed-running distance (km)	478	2.62	0.44	1.5	4.2
High-speed-running effort	478	188	32	103	303
High-intensity-running distance (km)	478	4.73	0.82	2.8	7.2
High-intensity-running effort	478	287	48	164	434
**Predictor variables**					
Temperature (°C)	472	21.4	6.4	2	34
Humidity (%)	472	66	20	12	100
AQI	478	79	56	18	500

Variables without Units Represent Counts.

**Table 3 ijerph-16-04238-t003:** Effects of temperature on the technical and physical performance of CSL teams.

Variables	Quadratic Effect	Linear Effect
	Optimum Temperature; ±90%CL	Standardized Effect; ±90%CL
Shot	18; ±12	
Shot on target		0.00; ±0.16 ^000^
Possession		−0.02; ±0.16 ^000^
Possession in opponent half		0.06; ±0.16 ^00^
Pass		0.03; ±0.16 ^000^
Pass accuracy		0.27; ±0.16 **
Forward pass	17; ±10	
Forward pass accuracy		0.27; ±0.16 **
Opponent 35 m entry		0.10; ±0.16 ^00^
Opponent penalty area entry		0.06; ±0.16 ^000^
Cross		−0.21; ±0.17 *
Corner		−0.01; ±0.17 ^00^
Offside	22; ±13	
50-50 challenge won		0.02; ±0.17 ^00^
Foul committed	13; ±18	
Yellow card		−0.15; ±0.17 ^0^
Red card		−0.16; ±0.17 ^0^
Total distance	11.6; ±4.7	
Sprinting distance	15.1; ±2.7	
Sprinting effort	13.2; ±3.8	
High-speed-running distance	12.0; ±3.5	
High-speed-running effort	10.6; ±4.3	
High-intensity-running distance	13.6; ±2.6	
High-intensity-running effort	11.6; ±3.7	

Likelihood of clear substantial effect: * possibly, ** likely. Likelihood of clear trivial effect: ^0^ possibly, ^00^ likely, ^000^ very likely.

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
