# Peer review of "Match Performance of Soccer Teams in the Chinese Super League—Effects of Situational and Environmental Factors"

_ijerph, 2019, doi:10.3390/ijerph16214238_

Round 1

Reviewer 1 Report

Check for tenses throughout the article.Page one, line 33: statues/results.....the word might be "match status" Edit for clarity, page one, line 34-38: perform higher numbers in goal scoring, passing and organising related technical actions, while committing fewer fouls and receiving fewer cards than playing away....Players from successful teams were generally more involved in actions with the ball possession [3,4], and covered more distance and high-speed-running distance whilst in ball possession Page 2, line 91: Ethics committee approval of this study was gained from the local university.....I suggest the authors specify the University from which the approval was secured. Transparency is vital here. Page 4, lines 126 - 128:  These lines need to be deleted as they might have been from the template for writing the article:  "This section may be divided by subheadings. It should provide a concise and precise description of the experimental results, their interpretation as well as the experimental conclusions that can be drawn". Replace it with a brief introduction/paragraph for the results section. Page 6. line 167-168: Edit for clarity regarding grammar: The CSL teams made most shot, forward pass, offside and foul committed  Page 7, line 178:  Edit "This study was aimed to identify the influence of situational" to read "This study aimed at identifying the influence of situational" Page 7, line 192-193: Edit  "believed to affect tactic and strategy choices in competition"  to read "believed to affect choice of tactics and strategy in competition"  Page 8, line 219: Instead of "reference team", use "stronger team" Page 8, line 233: Delete "will". Check for tenses throughout the article. Page 9, lines 251 - 252: Edit for clarity Page 9, line 254: Instead of "former", use "past" Page 9, line 257: "that is still required" and not "that it still is required" Page 9, lines 281 - 283: Edit for clarity Page 9, line 286-287: Our study demonstrated that situational variables had major effects on the technical performance but trivial effects on the physical performance of players in the CSL. 

Reviewer 2 Report

The present study is framed in an important line of research, increasingly emerging, as is the study of performance indicators in sport.

In this sense, researchers take another step and not only study the incidence of different contextual variables; analyze the incidence of environmental variables in sports performance (technical-tactical and physical)

They use an important sample of matches, and each one of them analyzes a multitude of technical-tactical and physical variables. They analyze three ambient variables. It is recommended to describe in more detail the environmental variables in particular describe more accurately the variable AQI (Air Pollution)

In the results section, because in table 2 the n (478) is repeated, it is recommended to remove it from the table and indicate it in the text

You should expand the information in the conclusions indicated that practical applications consider the author that should be provided (as for example in the World Cups when the temperature exceeds a threshold a break is made).

Reviewer 3 Report

GENERAL COMMENTS

The present study has merit despite lacks novelty.  The big data is the main strength of the article; however, information is missing about the validity and reliability of the instrument to collect the data. The article presents potential, but a major revision should be performed. Please consider my following specific comments per section.

ABSTRACT

Lines 16-23: please report the magnitudes of such increases or decreases. That will help readers to briefly understand the percentage of changes or how much they are different from the remaining conditions.

INTRODUCTION

Lines 42-52: I think that is missing here information about the acclimatization effect and the fitness status of the players. Certainly, that environmental factors will promote differences, however, the magnitude of such differences will be necessarily different considering the acclimatization of a player and his fitness level. I think that is something that should be extended here and certainly will be important to a stronger discussion of the results after.

Lines 53-61: It would be nice to read a related work about the mixing factor of two or more contextual factors interacting together to influence a given variable.

Lines 62-64: I think that it would be nice to make a summary of the main evidence found in the Chinese Super League using these references that are exclusive to this competition. This summary will allow understanding what is missing in the main research about the competition and will improve the statement of the contribution of this study.

Yang, G., Leicht, A. S., Lago, C., & Gómez, M. Á. (2018). Key team physical and technical performance indicators indicative of team quality in the soccer Chinese super league. Research in Sports Medicine26(2), 158-167. Lago-Peñas, C., Gómez-Ruano, M., & Yang, G. (2017). Styles of play in professional soccer: an approach of the Chinese Soccer Super League. International Journal of Performance Analysis in Sport17(6), 1073-1084. Brillinger, D. R. (2009). An analysis of Chinese Super League partial results. Science in China Series A: Mathematics52(6), 1139-1151. Mao, L., Peng, Z., Liu, H., & Gómez, M. A. (2016). Identifying keys to win in the Chinese professional soccer league. International Journal of Performance Analysis in Sport16(3), 935-947. Zhou, C., Zhang, S., Lorenzo Calvo, A., & Cui, Y. (2018). Chinese soccer association super league, 2012–2017: key performance indicators in balance games. International Journal of Performance Analysis in Sport18(4), 645-656. Gai, Y., Leicht, A. S., Lago, C., & Gómez, M. Á. (2019). Physical and technical differences between domestic and foreign soccer players according to playing positions in the China Super League. Research in Sports Medicine27(3), 314-325.

MATERIALS AND METHODS

Line 77. I think that a split in this section would help to follow the main idea. Maybe a sub-section of “sample” (e.g., characteristics, inclusion/exclusion criteria) and another one of “experimental approach” (e.g., study design, variables) would be better.

Lines 78-81: being discussed or used is not equal to “proven”. The references 16 and 17 used to justify the “accuracy, validity and reliability” of the system did not study or revealed any of these words/concepts about the instrument. The reference 16 (Andrzejewski et al., 2014) compared the physical and technical activities of professional soccer players using the system but not testing the system. The reference 17 (Di Salvo et al., 2007) just compared the physical demands between players and not tested the system. The authors should find and use appropriate references to justify such a strong statement. In fact, an absence of the validity and reliability of the instrument will compromise the confidence in the results presented.

Line 89: It would be better to describe how is calculated the air quality index and what represents. This is simple to interpret for temperature and relative humidity.

Line 91: please add the code number and the institution that provided the ethical approval

Line 93: how the opponent’s 35m entry is measured/coded? The system uses the player’s location in the Cartesian plan or is just using the observation?

Line 93: how 50-50 challenge is measured? What about a player that just partially delayed the play and passed this is not a 50-50 challenge? Please describe what means and detail the criteria

RESULTS

Lines 144 and 160 (figures 1 and 2): the concept of the figure is good but the definition/resolution not. Please try to increase the overall quality of the figure. Moreover, the title of the “x” axis is missing.

Possibly, would be interesting to test the interaction between contextual factors. The split analysis does not provide us a big picture of the context and how the variables interact together to explain the reality.  

DISCUSSION

Lines 228-266: information about the interaction between acclimatization and fitness status to explain the results or to highlight the limitations of the current study is missing. I think that just compare results between different levels of temperature or relative humidity too simplistic and did not consider the physiological adaptations of the players and a set of interactional variables that may provide a real understanding of the impact of these factors on player’s performance.

Line 284: two new paragraphs should be added: study limitations/future studies and practical applications.

Round 2

Reviewer 3 Report

The article was improved based on the reviewer's comments. Few final changes should be done and this the reason for my acceptance after minor revisions:

- The details about the opponent’s 35m entry and how 50-50 challenge is measured (that authors answered and detailed in the response to reviewer letter) should be added in the article (to clarify the readers and just not only us as reviewers). 

Author Response

Comments

The article was improved based on the reviewer's comments. Few final changes should be done and this the reason for my acceptance after minor revisions:

The details about the opponent’s 35m entry and how 50-50 challenge is measured (that authors answered and detailed in the response to reviewer letter) should be added in the article (to clarify the readers and just not only us as reviewers).

Thanks for your positive feedback, the details of two variables have been added. ( Page 4, line 117)